# Involvement of Gut Microbiota in SLE and Lupus Nephritis

**DOI:** 10.3390/biomedicines11030653

**Published:** 2023-02-21

**Authors:** Rozita Mohd, Siok-Fong Chin, Syahrul Sazliyana Shaharir, Qin Shi Cham

**Affiliations:** 1UKM Medical Centre, Department of Medicine, Faculty of Medicine, Universiti Kebangsaan Malaysia, Jalan Yaacob Latif, Cheras, Kuala Lumpur 56000, Malaysia; 2UKM Medical Molecular Biology Institute (UMBI), Universiti Kebangsaan Malaysia, Jalan Yaacob Latiff, Cheras, W. Persekutuan, Kuala Lumpur 56000, Malaysia

**Keywords:** gut microbiota, SLE, lupus nephritis, leaky gut, probiotic

## Abstract

Lupus nephritis is a severe manifestation of systemic lupus erythematosus (SLE). It is caused by immune dysregulation and kidney inflammation. In recent findings, gut microbiota potentially acts as primary mediators to enhance immune complex deposition, complement activation, and macrophage infiltration, and led to renal inflammation. Gut inflammation, known as leaky gut, allows pathogenic bacteria to enter the blood stream to form immune complexes which deposit on the kidney. Lymphocytes and macrophages induct a proinflammatory cytokine milieu that leads to kidney inflammation. Accumulating pieces of evidence from the field of gender bias, dietary habit, alcohol, smoking and antibiotic consumption were closely related to dysbiosis of gut microbiota in SLE. However, little is known about the causes of gut microbiota dysbiosis and the potential pathway that leads to lupus nephritis (LN) flare. In this review, we will bring into deeper insight for the potential link of gut microbiota on immune system with a particular focus on renal inflammation. Moreover, we also discuss the potential novel therapies that regulate gut composition to improve or complement the current treatment of LN.

## 1. Introduction

Systemic Lupus Erythematosus (SLE) is an autoimmune disease resulting in multi-organ inflammation with complex clinical manifestation including neuropsychiatric, lupus nephritis and so on. However, the actual cause of SLE is still unknown [1]. Various aetiologies have been postulated which can be categorized into genetic and environmental such as age, gender, ultra-violet exposure, dietary intake, alcohol, and lifestyle behaviour [2]. Lupus nephritis is one of the most severe manifestations of SLE with high mortality rate and 10% of them eventually develop end-stage renal disease within 5 years after diagnosis [3,4].

In a review of medical records of 258 SLE patients in our centre from 2007 to 2017, majority of the SLE patients were Malay (56.2%) followed by Chinese (38.8%) and Indians (5%) [5]. Renal involvement in our cohort is high, affecting up to 60% of the patients [6]. Most importantly, patients with LN were found to be at risk of organ damage as almost half of them accrued damage in the duration of follow up of up to 20 years. Although SLE is less common among the Indian population, they are at risk of developing LN (61.1%) [7] and disease damage (66.7%) [6]. Therefore, racial diversity in Malaysia has different organ and system involvement.

Recently, enormous reports highlighted on the association of autoimmune diseases with dysbiosis of gut microbiota [8,9,10,11]. There are trillion of microbes inhabited human gut which dominated by 4 phylum such as Firmicutes, Bacteroidetes, Proteobacteria and Actinobacteria. Every species of gut microbiota plays their own role in modulating immune system, absorption of metabolites, short-chain fatty acid (SCFA) metabolism, and so on. Many factors could alter gut communities such as infant transition, dietary habit, age, gender and antibiotic consumption [12]. Dysbiosis of gut microbiota will affect the intestine membrane permeability, tight junction, mucosal barrier and intestinal metabolites composition. At present, the role of gut microbiota in LN pathogenesis are less understood. In a few reviews paper, gut dysbiosis has been shown to be associated with SLE while little is known on lupus nephritis [13,14,15].

In this narrative review, we will discuss about the associate factors of gut microbiota in SLE and LN and the specific microbial species that possibly link to flare of LN. We hope results from recent experimental studies could provide new insight for future human study to determine how gut microbiota affects lupus nephritis.

## 2. SLE and LN Pathogenesis

B cell dysfunction is still considered as the key factor for immune dysregulation resulting in autoantibodies production in SLE [16]. Other patho-mechanisms of SLE are defective in apoptosis due to ineffective clearance of cell debris during cell death mechanism which exposes nuclear material (nuclear protein, cytoplasmic protein and membrane components) to antigen presenting cells (APCs) [17,18]. Upon the nucleic material being picked up by the APCs, IFN-a was produced and stimulate IL-6 production which [19] act as an immune adjuvant in the maturation of T helper cell [20]. The hyperactivation of T-helper cells will then activate B cells to initiate plasma cells antibodies production. These circulatory immune complexes deposit on kidneys and activate the complementary reaction which led to kidney inflammation, named as lupus nephritis (LN).

## 3. Gut Microbiota Dysbiosis of SLE

Gut microbiota as gut commensals in host gastrointestinal tract especially significant in immune modulatory. During the autoimmune disease condition, the symbiotic relationship is broken due to various factor such as dietary habit that change microbiota diversity. The decrease of microbial diversity was found in autoimmune diseases such as SLE and inflammatory bowel disease (IBD), presented by reduce commensal bacteria (Firmicutes and Bacteroidetes) and increase of detrimental bacteria (Proteobacteria and Actinobacteria) [21].

Homeostasis of gut microbiota maintain the function of dietary metabolism of the general host, formation of protective barrier, tight junction expression and immunology regulation [22]. Microbiota involves in dietary metabolism of digested protein, carbohydrate and dietary fibre. There are 100 trillion bacteria that form gut microbial community in general human population, where the composition could be easily influenced by age, gender, genotype and food intake [23,24]. In healthy human gut, the majority of the bacteria (98%) is dominated by two major phylum which are Bacteroidetes and Firmicutes [25]. Firmicutes responsive in fatty acids and carbohydrates absorption whereas Bacteroidetes correspond to polysaccharides absorption [26]. Alteration of gut microbiota expresses as Firmicutes/Bacteroidetes ratio potentially act as the measurement for pathological condition in SLE patients which cause systemic inflammation. The decreased ratio of F/B microbiome in SLE patients occur either by the reduction in Firmicutes or increase abundance of Bacteroidetes. The lower F/B ratio in SLE patients was proven by a Spanish cross-sectional study, which compared gut microbiota of SLE patients in remission with sex-matched health control [27]. However, the gut microbiota was found to be reduced in microbial diversity and complexity in lupus patients and lupus-like mice [28,29]. Furthermore, SCFA (butyrate) and tryptophan, as indispensable role in activating B cell and generate autoantibodies which initiate autoimmune intolerance [29,30]. In human study, SCFA was found in fecal level of lupus patient with low Firmicutes to Bacteroidetes (F/B) ratio suggesting the potential link of gut absorption function with microbes [31].

However, the effect of F/B ratio also show significant in other autoimmune disease such as IBD and rheumatoid arthritis [32]. Hence, it is necessary to identify the exact mechanism on how the gut microbiota promote SLE and LN pathogenesis in human studies.

### 3.1. Dietary Habits

Transgenic mice TLR7.1 Tg mice was closely resemble to the TLR7-pDC-IFN axis in SLE patients. In TLR7.1 Tg mice study suggest resistant starch (RS) rich diet producing SCFA, suppress the enrichment of *L. reuteri*, by decreased pDCs and interferon pathways to reduce lupus symptoms [33]. Hence, the evidence manifest that RS rich diet could suppress interferon pathway from gut pathobiont which particularly important in human SLE pathogenesis. Interestingly, high level of fecal kynurenic acid as one of the common tryptophan metabolites, have been identified in SLE patients from a China report [34]. High tryptophan diet exerts proinflammatory effect to triple congenic (TC) lupus-prone mice which exhibit altered microbial communities with elevated level of *Lactobacillus* spp. and *Bacteroides dorei*. In contrast, low tryptophan diet in TC mice improved lupus feature such as Tregs functionality which was similar to germ-free B6 mice [30]. The expression of CD25 on Treg cells was increased in TC mice after low tryptophan diet while high tryptophan diet demonstrate increased expression on Th 17, B cell and plasma cells [30]. From the findings above, gut microbiota could initiate immune response in SLE through tryptophan degradation.

Hence, in order to shape immune tolerance gut environment, dietary habits such as low tryptophan, high fibre and resistant rich diet were essential to maintain normal gut function. Metabolite profile can be the potential measurement on microbial activity in mediate SLE.

### 3.2. Gender

Despite female preponderance of 9:1 ratio, many studies showed that males with renal involvement have higher mortality rate and more severe disease manifestation [2,35]. Gender bias was a significant factor in affecting predominant of SLE and severity of LN. SLE affects predominant woman of childbearing age indicating hormonal factor could be the predisposing factor for SLE.

Immunoregulatory function of oestrogen were range from various immune cells of innate immunity to adaptive immunity [36]. The IFN-stimulated genes were highly stimulated to secrete proinflammatory cytokines after in vivo treatment of 17β-oestradiol with PBMCs from SLE patients and healthy controls [37]. Furthermore, oestrogen was related to renal disease exacerbation. The female MRL/lpr mice develop severe glomerular damage after consumption of a synthetic 17β-oestradiol, which was 17α-ethinyl oestradiol [38].

Higher oestradiol and lower plasma testosterone in female reduce T-regulatory cells (Tregs) ability to express FoxP3 which is postulated to cause SLE [39]. The protective effect from male hormone associated with gut microbiota generally inhibit disease development in early life of male BWF1 lupus model [40]. Interestingly, the gut microbiota was also significantly different in both genders. Early development of lupus in the female and castrated males BWF1 mice strongly suggest the influence of sex hormones. However, male patients were more prevalent in developing higher disease activity and renal related autoimmune disease [41]. Whether composition gut microbiota influence this effect is still unknown.

### 3.3. Geographical Factor

SLE patients in Northeast China were found to have different gut microbiota profiles compared to Spain and a Southern China cohort. The intestinal microflora of Northeast China was significantly increased in phylum Proteobacteria, family *Enterobacterlaceae* and *family_XI_o_Clostridiales* and decreased in family *Prevotellaceae*. On the other hand, Spain and Southern China patients did not have any changes in family *Enterobacterlaceae* and *family_XI_o_Clostridiales*. Although the exact mechanism to the changes of microflora in Northeast China still need deeper exploration, we cannot deny the possibility of gut microbes in SLE patients were influenced by geographical regions [42]. Analysis from 1995 to 2010 in 41 study centres showed that prevalence of LN patients increased with decreasing latitude from the north to the south part of China [43]. More research is needed to explain the relationship between geographical factors with gut microbiota of LN and SLE patients.

### 3.4. Concomitant Medication

Kidney involvement affects 60% of SLE patients which require treatment with non-specific immunosuppressive such as glucocorticoid. The effect of glucocorticoid on gut microbiota was shown by M. Guo et al., 2020. In this study, the gut microbiota of two human SLE groups (with or without glucocorticoid) and healthy human control were compared. Glucocorticoid treatment had almost similar gut microbial community with healthy human control [44]. They have increased Firmicutes to Bacteroidetes ratio indicating enrichment of Firmicutes and depletion of Bacteroidetes, including *Lactococcus, Streptococcus*, and *Bifidobacterium.* Those genera activate anti-inflammatory mechanism such as glycan metabolism, SCFA production, tight junction and mucosal stimulation thus restoring the gut function of SLE patients. Unfortunately the glucocorticoid has adverse effects for SLE patients on cardiovascular, optical disease, psychiatric disorders [44] and increase risk of infection [45]. Treatment with Hydroxychloroquine (HCQ) was found to reduce infection risk as shown in C57BL/6J mice [45]. The ratio for Firmicutes/Bacteroidetes decrease in C57BL/6J mice after 7 days high dose of HCQ suggesting the inflammatory influenced by alteration of gut diversity and gut membrane integrity [46].

## 4. Potential Initiation of LN by Gut Microbiota

### 4.1. Microbes Metabolite Host Axis

There were numerous studies establishing the effect of certain types of diet with gut microbiota species in lupus nephritis. Diet with low fibre promotes immune dysregulation resulting in deposition of immune complex in kidney glomeruli as observed in the control C57BL/6J mice. In contrast, the lupus symptoms were significantly attenuated in mice lacking Lyn tyrosine kinase (Lyn−/− mice) that were fed with high fibre diet. The link between high fibre diet and immune modulatory in mice could suggest a similar possibility in humans as well [47]. The non-digestible carbohydrate, fibre will be fermented into SCFA [33]. Most of the SCFA exist in gut are acetate, propionate and butyrate [48]. Immunoregulatory function of SCFA were range from anti-inflammatory, T cell and epigenetic pathway. *Bacteroides* and *Negativicutes* produce propionate through the succinate pathway [49]. However, butyrate was mainly produced by phylum Firmicutes via acetate CoA-transferase pathway [49]. Dysbiosis of gut microbes in SLE was associated with the SCFAs production by increase of SCFAs in fecal level and reduce F/B ratio [48]. However, butyrate has been demonstrated to attenuate kidney disease via G-protein coupled receptor [50]. Butyrate was essential to maintain gut homeostasis by activate PPAR-γ-signalling, consequently suppress inducible nitric oxide synthase (iNOS) synthesis and prevent overgrowth of nitrate dependent microbes in gut [30]. Study using butyrate treatment in lupus-prone mice showed it could ameliorate kidney damage by increasing F/B ratio and microbial diversity [29]. In NZB/W mice, treatment with histone deacetylate 6 inhibitor showed significant reduction in lupus nephritis symptoms by inhibiting B cell activation pathway [51]. Therefore, the significant relationship between SCFA and histone deacetylate (HDAC) provide insight of SCFA as an immunomodulator for LN. Hence, microbial metabolite was significant in immune regulation between diet and gut microbiota [50].

### 4.2. Sex Microbes Axis

The link of sex hormones, such as oestrogen and androgen with X and Y chromosome can be used to explain sex bias on disease predominant and severity. Oestrogen associates in female upregulated X-linked genes, ERα, preferably activate Th2 immune response and autoreactive B cell [41,52]. In a study of TGP treatment on SLE mice, ERα in B cell contribute to glomerulonephritis while TGP treatment on SLE mice able to reverse the pathogenic condition [52]. However, there were higher chance for a men to develop chronic lupus nephritis with androgen through programmed cell death pathway in kidney [41]. The anti-dsDNA antibodies in male mice was elevated with the presence of Y chromosome with Tlr7 gene expression and develop LN [53]. However, upregulation of the related epigenetic was due to gut dysbiosis [54].

Female lupus mice were found to have higher levels of *Lachnospiraceae* whereas male lupus mice have higher *Bifidobacterium* in the gut [55]. In addition, *Lactobacillus* treatment resulted in significant improvement in the renal inflammation and worked only in female and castrated male lupus mice [4]. Post *Lactobacillus* treatment, castrated lupus mice showed significant decrease in IgG2a and IgA while increase TGFβ and IL-10, which reduce the LN pathology. These findings could suggest the correlation between different gut microbiota with different gender and hormonal profiles with immune responses. Level of serum creatinine and haematuria was significantly higher in male SLE patients compared to female SLE patients of similar age, race and disease durations [56].

Hence, female contain higher risk in SLE when compare with males. In addition, males consist of higher risk in renal damage.

## 5. Clinical Significance of Gut Microbiota in Lupus Nephritis

### 5.1. Leaky Gut in Lupus Nephritis

Dysbiosis of gut microbiota causes impairment of intestinal barrier function which lead to increase intestinal permeability to certain pathogens, viruses, and pathogenic antigens into internal organs. Figure 1 summarized the associated gut microbiota in signalling kidney inflammation. Defective of barrier function was observed in blood stream of lupus prone lpr mice in the presence of endotoxin lipopolysaccharides (LPS) [4]. The gut damage in lupus mice is caused by the increase in LPS which is the major component of cell wall of gram-negative bacteria [57]. LPS cause disruption of tight junction proteins in lupus gut such as ZO-1, occluding and claudins [58]. It could also stimulate nuclear factor κB which is critical for LN pathogenesis, alteration of proinflammatory cytokine milieu and mediate tubulointerstitial injury in renal [59,60,61]. Exposure of LPS in nephritic mice enhanced the production of DNA autoantibodies, IL-6, IL-12, TNF and interferon (IFN)-β level which express TLR2 on glomerular endothelial cells [62]. High albumin excretion occurred in LN when the permeability of albumin increased in glomerular endothelial cells upon the activation of LPS [62]. On the other hand, colonization of *Lactobacillus* and *L. reuteri* in MRL/lpr mice promote LPS clearance via stimulation of Intestinal Alkaline Phosphatase (IAP) hence strengthening the tight junction and barrier function. IAP will inhibit activation of NF-kB pathway by preventing the binding of LPS through TLR 4 in releasing proinflammatory cytokines [63]. Treatment with SCFA as HDAC inhibitor has been proven in preventing the activation of NF-κB pathway in experimental lupus mice [51]. The weakened tight junction increases permeability of gut barrier promoting further leaky gut effect. Gut leakage contributes to anti-RG autoantibodies which react to RG antigen was found in higher abundance and aggravate renal involvement in lupus patients [64]. In addition, RG species was found in the faecal of 4/9 of LN and 1/7 of non-renal patients in a cross-sectional of SLE human study [65]. Hence, the elevated of relative abundance of RG was strongly related to lupus especially in renal pathology [64].

### 5.2. Molecular Mimicry

The cross reactivity between certain bacteria antigen with specific autoantibodies has been recognized as the main factor of leaky gut syndrome in lupus. Pathogenic bacteria were detected in lupus gut with bacterial ribonucleoprotein ortholog induce production of anti-Ro60 antibodies. The relationship of bacterial infection and kidney inflammation was reported in various human and animal study, specific species including of *R.torques* spp., *Ruminococcus gnavus*, *Staphylococcus aureus*, *Bacteroides thetaiotaomicron* and *Streptococcus pneumoniae*. There were significant data that indicated the shift of gut microbiota in lupus nephritis (Table 1). Colonization of *Bacteroides thetaiotaomicron* in lupus mice was found to enhance the expression of Ro60 antigen followed by deposition of immune complex causing lupus nephritis [66]. Meanwhile, *Staphylococcus* was found in high abundance especially in middle age to older lupus patients. *Staphylococcal* antigen generates immune complexes with consequence glomerular deposition, with 44% of lupus patients were eventually end-stage renal disease [67]. Other causes that probably contribute to renal inflammation include of anti-HU1 antibodies that cross react with self-antigen, P4HB on renal cells for SLE patients and pristane-induced SLE murine model in vivo. The anti-HU1 antibodies was stimulated by DNABII proteins that present in biofilm of *S. aureus* [68].

### 5.3. SFB Colonization

Valiente et al. (2022) further proved the correlation between gut dysbiosis with Segmented Filamentous Bacteria (SFB) in LN [14]. The SFB colonized mice (NZM2410) experienced gut dysbiosis at week 15 to 30 and experienced severe glomerulonephritis stages at week 30 as compared to Germ-Free mice. Furthermore, renal related symptom closely related when the increment of MCP-1, CXCL1 and kidney infiltrating F4/80+CD206+M2-like macrophages in lupus mice. *R. torques* spp. was found higher abundance in +SFB at week 30. Similar observation seen with *Ruminococcus gnavus* which share the same ancestor with *R. torques* sp. that elevated in human LN. *Ruminococcus gnavus* within *Lachnospiraceae* family was overrepresented in initiating flare of LN especially with high SLEDAI score [64].

### 5.4. Th17/Treg Balance

Renal inflammatory response achieved in 60% of SLE patients. In experimental model, the renal responses in NZBWF1 mice can be identified through urinary creatinine, albumin and blood pressure that associate with anti-dsDNA and LPS levels. As previously mentioned, the barrier integrity of intestinal epithelial in lupus mice can be restored by bacteria *Lactobacillus fermentum CECT5716* (LC40) leading to inhibition of deposition of immune complexes in kidney. LC40 treated mice for optimum Th17/Treg ratio to inhibit accumulation of proinflammatory cytokine, IFN-α and reduce B cell activation [69]. The positive feedback of LC40 toward renal function suggest that probiotic LC40 can be utilize as a supplement to improve LN treatment.

### 5.5. Role of IgA

Symbiotic balance of intestinal microbiota influenced by Immunoglobulin A (IgA) contribute to LN onset and progression. In intestinal mucosa, IgA induced by specific gut microbes and synthesis in B cells via dendritic cells activation [70]. IgA function as bacterial or toxins neutralizer to block the adherence and entry of pathogenic bacterial or viruses on intestinal mucosal surface [70,71]. The production of IgA was directly proportional to Tregs which is crucial for disease pathogenesis regulation [71]. When Tregs was disrupted, it will inhibit the production of IgA which aggravate lupus pathogenesis [71]. Moreover, IgA have proven to be associated with kidney severity in SLE patients through the observation of paper from January 2009 to June 2012 [72].

## 6. Potential Treatment for Lupus Nephritis

The mainstay therapy for lupus nephritis is glucocorticoids, steroid sparing agents and also non-steroidal anti-inflammatory drugs (NSAIDs) that widely used for NF-κB downregulation. However only 60% to 70% of LN patients achieve complete or partial remission. There is a summary of potential treatment in lupus nephritis on Table 2.

### 6.1. Probiotic Treatment

In experimental study, certain bacteria strain able to ameliorate the pathogenic symptom of autoimmune disease such as *L. rhamnosus JB-1*, *B. breve yakult*, *L. paracasei*, *L. plantarum*, *L. delbrueckii*, *B. lactis*, *L. reuteri* and *B. longum* [73]. In addition, LC40 able to restore the immune integrity by regulate Th/Tregs cell ratio, thus improve renal response in lupus mice [69]. Besides probiotic treatment, the treatment duration or time-point was crucial in lupus severity. The influence of early and short-term interventions of gut microbiota have been proposed toward genetically caused lupus in animal study [74]. The short-term changes of gut microbiota with faecal microbiota transplantation (FMT) may impose long term effects in suppressing disease progression for MRL/lpr mice. Furthermore, short-term treatment of FMT before disease onset was found to improve the glucocorticoid efficacious [74].

However, the long-term efficacy of FMT treatment toward lupus patient have yet to be determined. The mammalian evolution was affected by gut microbiota, which may induce evolutionary dependence [75]. Furthermore, the efficacy of short term FMT treatment may influence by within-host evolution of a gut pathobiont such as *E. gallinarum*, *B. fragilis* and *L. reuteri*. According to In-vivo experiment, *E. gallinarum* was evolve into adapted lineage with greater translocation ability, hence aggravate liver inflammation in germ-free (GF) mice model [76].

### 6.2. Tuftsin-Phosphorylcholine (TPC) Treatment

TPC is well known on treatment of helminths parasite and modulation of immune response [77]. However, it was also found to modulate microflora species of the gut and suppressing the proinflammatory cytokine related in T cell activation. TPC administration was effective in reducing IL-1β and IL-6 and inducing IL-10 for anti-inflammatory purpose hence ameliorate nephritis symptoms [78]. Similar result was observed in TPC treated mice with active lupus causing significant reduction in proteinuria level which associate with the production of cytokine to skewed into Tregs production. Pathological effect can be reversed after 26 weeks of TPC treatment especially with elevation of protective microbes such as *Bifidobacterium* and *Adlercreutzia*. TPC treated mice promote more significant result than butyrate treated mice as Bifidobacterium contribute for better metabolism to butyrate. Hence, TPC enhance the protective effect of butyrate in maintenance of mucosal layer in the gut intestine which normally weaker in lupus condition [77].

### 6.3. Butyrate Supplementary Treatment

Butyrate is a metabolic product of intestinal microbes that influence to immune response by increase systemic inflammation [79]. Dietary butyrate playing important role in antibody regulation by maintain microbial dysbiosis and balancing of Treg and TFH cells in Rheumatoid Arthritis (RA) which is similar to SLE and LN [80]. The production of short-chain fatty acids can be enhanced through high-fibre dietary intervention [81]. According to a short term dietary study with high fibre dietary supplementation, RA patients exhibit higher Treg numbers and reduce disease progression [82]. Moreover, there is another evidence about the beneficial of short-term dietary fibre intervention in RA patients by increase the level of SCFA, reduce proinflammatory cytokine and increase Firmicutes/Bacteroidetes ratio [83]. Dietary interventions in RA patients were using Gum Arabic (GA) which mainly fermented into butyrate and propionate [84]. As a consequence, the significant reduction in urea and liver enzyme was found on RA patients during the phase 2 trial with GA which enhance the protective effect on liver and kidney part [84].

### 6.4. Treatment for Leaky Gut Symptom for IBD

Leaky gut is a common symptom for autoimmune disease during depletion of tight junction protein on intestinal membrane such as ZO-1 and ZO-2 [85]. Gut leakage have been correlated with the renal inflammation in lupus patients when bacterial antigen cross reacts with autoantibodies. Although the possible treatment of leaky gut in SLE and LN have not yet been well identified to date, the treatment of leaky gut in IBD can be related closely with SLE and LN. From an experimental study involve of three IBD biomimetic systems, the leaky gut symptom can be restored by schisandrin C which is an active compound from *S. chinensis.* The expression of tight junction protein can be induced by schisandrin C that crucial for intestinal barrier function [86]. Furthermore, probiotic such as *Bifidobacterium longum NK219*, *Lactococcus lactis NK209*, and *Lactobacillus rhamnosus NK210* enhance tight junction protein, suppress the LPS production from gut bacteria and subsequently alleviate leaky gut condition in gut inflamed mice [87]. Hence, probiotic treatment with *Bifidobacterium longum NK219*, *Lactococcus lactis NK209*, and *Lactobacillus rhamnosus NK210* are potential in leaky gut impairment of LN.

**Table 2 biomedicines-11-00653-t002:** The potential treatment of Lupus Nephritis in animal study.

Study Types	Method	Study Group	Results	Reference
*Animal Experimental Study*	Chronic oral administration of Probiotic	SLE Mice	Decrease in renal oxidative stress and inflammation Reduced plasma levels of autoantibodies and LPS.	[69]
*Animal Experimental Study*	Tuftsin-phosphorylcholine treatment	SLE Mice	Increases CD1d expression level and reduces CD86 expression level Reduce *Akkermansia muciniphila*, genera *Clostridium*, *Anaerostipes*, and *Anaerotruncus* Increase *Bifidobacterium* and *Adlercreutzia*	[77]
*Animal Experimental Study*	Short-term FMT treatment	SLE Mice	Suppress lupus progression (reduce anti-double-stranded DNA, increase interferon-alpha)	[74]
*Human Study*	Dietary Butyrate treatment	RA Patient	Reduce Urea enzyme, liver enzyme	[84]
*Animal Study*	Probiotic treatment	IBD Mice	Increase colonic interleukin-10 expression and mucin protein-2+ cell and claudin-1+ cell numbers expression Reduce tumor necrosis factor alpha and interleukin-1 βeta expression, NF-κB+Iba1+ cell and LPS+Iba1+ cell numbers in the hippocampus	[87]

CD: cluster of differentiation, SLE: systemic lupus erythematosus, RA: rheumatologic Arthritis, IBD: inflammatory bowel disease, LPS: lipopolysacharride, NF-κB: nuclear factor kappa beta, Iba1: ionized calcium binding adaptor molecule 1.

## 7. Challenges and Future Outlook

This review highlights developing literature to prevent renal inflammation by modulating gut microbiota composition. LN pathogenesis pathway can be observed in both animal and human studies in regards to gut membrane infection, NF-κB pathway, immune complexes formation and proinflammatory cytokine production. Gut dysfunction in both human and mice was strongly correlated to renal lupus. The relationship of human and mice can be observed in *R. gnavus* and *R. torques*, respectively, with similar ancestor, while both elevated during LN [14,64]. The renal disease in SLE was demonstrated with the higher level of autoantibodies such as the anti-RG antibodies which against to the antigens in *R. gnavus* lipoglycan [64].

There are few studies suggest on the short-term effectiveness of *B. fragilis* and *Lactobacillus fermentum CECT5716* as probiotic for animal with renal inflammation [69,88]. However, the effectiveness of *Lactobacillus* treatment in LN patients still yet to be proven. More human research especially on LN patients is required to ensure the effectiveness of probiotic to improve or replace current LN therapy.

Early prognosis and diagnosis are especially critical for LN disease manipulation. Hence, timepoint for probiotic treatment should be another attention on the effectiveness for long term LN remission. Probiotic still yet to be proven for replacement of current LN therapy, there is another insight on reduction in long-term medication side effect.

## 8. Conclusions

In summary, understanding the altered gut microbiota composition and their role in disease manifestation could suggest new therapeutic strategy for SLE and LN. Growing evidence from SLE and LN studies on altered gut microbiota composition that possible influence by diet, gender, geographical and medication. The metabolite profile can be the potential measurement on microbial activity in mediate autoimmunity. Gut microbiotas have significant effects on metabolic pathway of dietary components. Sex hormones affect the gut microbiota that result in female predominance in SLE while male have greater severity in renal impairment. Overexpression of specific gut commensal help us predict LN pathogenesis. However, most of the reported studies were from animal models which lacking an in vitro or ex vivo on human study. Recently, there were few animal studies provide an insight on effectiveness of probiotic and TPC treatment. However, several treatments for parallel disease in RA and IBD were potential for treatment of SLE and LN. The main limitation in this review is lacking on the data from human research. Hence, more research or clinical trials on human gut microbes related with LN flare is urgently required to advance our knowledge for better control of LN disease activity.

## Figures and Tables

**Figure 1 biomedicines-11-00653-f001:**
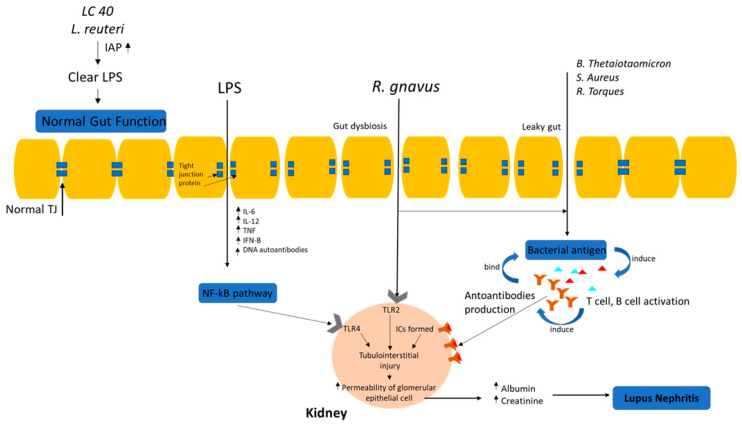
Summary of defective barrier function caused by gut dysbiosis that promote to lupus nephritis. IAP: intestinal alkaline phosphatase, LPS: lipopolysaccharides, TJ: tight junction, IL: interleukin, TNF: tumor necrosis factor, IFN: interferon, NF-κB: nuclear factor kappa beta, TLR: toll-like receptor, IC: immune complexes, 

: Increment.

**Table 1 biomedicines-11-00653-t001:** The shift of microbiota in Lupus Nephritis for human and mice.

Microbial Species	Changes	Study Group	Effects	Reference
*R. torques* spp.	Increase	Mice	Increase of MCP-1, CXCL1 and kidney infiltrating F4/80+CD206+M2-like macrophages	[14]
*Ruminococcus gnavus*	Increase	Human	High anti-RG antibodies, high SLEDAI score, high antinative DNA levels, but low complement 3 and complement 4 level	[64]
*Staphylococcus aureus*	Increase	Human	High anti-HU1 antibodies	[68]
*Bacteroides thetaiotaomicron*	Increase	Mice	High anti-Ro60 antibodies	[66]
*Lactobacillus fermentum CECT5716*	Increase	Mice	Low anti-double stranded DNA antibodies, T, B, Tregs and T-helper 1 cells	[69]

MCP-1: monocyte chemoattractant protein-1, CXCL1: chemokine ligand 1, RG: Ruminococcus gnavus, SLEDAI: Systemic Lupus Erythematosus Disease Activity Index.

## Data Availability

Not applicable.

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
