# Peer review of "Involvement of Gut Microbiota in SLE and Lupus Nephritis"

_biomedicines, 2023, doi:10.3390/biomedicines11030653_

Round 1

Reviewer 1 Report

General comment for authors

Overall, there are many typos and areas where the language is unclear throughout this review, making it difficult for the reader to follow. Comments/edits below are included via line item. It is an interesting review that highlights good points, but does require major revisions, expansion of certain topics, prior to appropriate publication for MDPI.

Grammatical Edits are included here:

15 – act as primary mediators
16- known as leaky gut, allows
17- the blood stream … on the kidney
18 – in renal? Flip sentence – lymphocytes and macrophages induct a proinglammatory --- mileau that leads to kidney inflammation
23 – cut the
24-25 – Potential novel therapies that regulate gut composition

29 – make more specific, and would be “clinical manifestations, including ______.”
34 – diagnosis
35-36- Font looks different
40 – among the Indian population
45- which is postulated

We did not cite every flaw nor provide edits for all of the grammatical areas throughout the manuscript – the editor can work on this further, but here we also highlighted a few additional typos.

75 – 100 trillion bacteria that form
77 – change in font again
81 – SLE patients
88 – incomplete sentence
93 – Human studies.
94 – dietary habits
98 – symptoms
114-  humans
117 - symptoms
119 – as an immunomodulator
125 – typo
148 – a Southern China cohort
157 – geographical factors
186 – plural receptors --- if just 1, which one?
330 – there are few studies suggesting short term effectiveness of  ------ probiotic supplementation as a tool for treating renal inflammation in mouse models? Which models? Include details of what are actually treated in these cases as well.
344 – effects
345-  that result in
348 – animal models
352- is urgently required

Content edit suggestions are included here:

40-41 – give numbers for how many (i.e., proportion of those with disease) develop LN and age compared to rest of population
44 – 55 - This should have a separate subtitle and be part of a larger separate section on etio-pathogenesis – it is specific into mechanisms without a header indicating as such. The 2 paragraphs before are general openers and then line 44 onwards should be a separate subsection
56 – Paragraph starting at line 56 can be part of the intro area, separate from the detailed pathogenesis section
73-74 – in introduction for gut dysbiosis, it should be made clear the difference between actual gut symbionts that then may drive lupus disease vs. actual pathogenic bacteria that drive active GI gastroenteritis and IBD. Active colitis, with features akin to IBD is not common in SLE itself. Should highlight an accepted term for symbiotic bacteria that colonize our gut, but may contribute to autoimmune pathogenesis can be considered “Pathobionts” distinct from gut disease pathogens. This should be highlight in a paragraph at the start of section 2.
91-93 – Expand further, highlight how the link to microbes themselves related to gut absorption vs. gut microbial metabolites related to changes in F/B ratio. What are the specific missing steps for why F/B ratio alone has not yet led to targetable outcomes in SLE – cite specific studies that assessed butyrate and tryptophan metabolisms (Need citation for line 88-89 too).
104 – Please include the actual details from the citation – what features of SLE and Tregs were improved in citation 34
143 – Need to have citation, please do not just say males always have more disease activity. Already noted 9:1 ratio In the background. This is likely unnecessary --- if want to talk about it need citation, with details and clear limitations with further steps for research linking gender disparities w gut microbiota 143-145
172 -  needs citation for HCQ and infection risk
193 – Maybe best to combine section 2.2 from line 125 down with 3.2 and it is not needed in 2.2 above esp since gender already discussed in the background. Alternatively put 3.2 up with 2.2 and cut from section 3. It is difficulty for reader to follow when separate as is now.
line 234 – is there data that shows SFB directly targets the kidney and then drives tubulointerstitial injury? If so, this needs to be emphasized and cited bc this is not shown in the text of the review, and therefore should not be highlighted as such in the figure. It can instead by included with other specific pathobionts. In this section relevant to sex-associated mechanisms of pathogenesis, can also mentionSex-dependent lupus RG strain Frontiers in Immunology (Aug 2022) PMID: 36032126 

.
Section 5 – line 288 – can also add for treatments parallels from other diseases, specifically treatments trialed in RA with the use of supplemental Butyrate and also treatments against the leaky gut used in IBD. Could these approaches be useful in SLE and LN as well? These considerations should be discussed
301 – Need to reference within-host evolution paper when considering if short term changes in microbiota can in fact cause long term changes, if instead a proposed theory is that within-host evolution of the pathobionts within our GI tract interact with our gut endothelium in susceptible individuals to select strains that then drive further autoimmune disease. If we only change gut microbiota at a single time, without also affecting our host propensity for autoimmune disease, will our gut pathobiont community again adapt to contribute to autoimmune disease? Whether or not long term effects can derive from probiotic cross sectional supplementation can be further discussed.

Line 323 – This review highlights developing literature – the review itself does not open opportunities

It should be mentioned that renal disease in SLE in part reflects immune complex disease, and antibody response to intestinal bacterial antigens, including R. gnavus lipoglycan, are likely contributory.

Figure edits.

Figure 1 indicates that LPS may be released from the gut when there is increased intestinal permeability, but indicates the wrong receptor, as this should be revised to indicate TLR4. Notably, a lipoglycan from strains of Ruminococcus gnavus has been shown to trigger immune activation through TLR2 Azzouz et al. (PMID: 30782585).  This should be revised in the figure.

Reviewer 2 Report

My comments: 

1. The whole paper is a litle bit confusing, it would have been better to discuss SLE and lupus nephritis separately or explane the same problem once, we can read about gender differences and hormonal effects in two separate parts of the review. 

2. in row 29: Why do you mention zhe year 1942, please delete it. 

3. in row 44. You write on the effects of estradiol on T regs, but it is known that it had an effect on other immun cells as well, please explane this in more detail. 

4. In row 53 and 326 the "proliferation" is not correct word. 

5. In row 306: " the use of helminths" really? 

6. You have got two table 1.

7. The references do not meet formal requirements. The exact references is incomplete. 

Please try to revise the manuscript. 

Round 2

Reviewer 2 Report

The revised manuscript is much better than the prevoius was. 

The referencees are still not precise and uniform.  

For example: 

Vol. 6, Frontiers in Immunology. 2016. 

Front Immunol. 2021;12

Please correct it. 

Author Response

Done. Thanks for your suggestion.